# CRISPR Therapeutics for Duchenne Muscular Dystrophy

**DOI:** 10.3390/ijms23031832

**Published:** 2022-02-06

**Authors:** Esra Erkut, Toshifumi Yokota

**Affiliations:** 1Department of Medical Genetics, Faculty of Medicine and Dentistry, University of Alberta, 8613-114 Street, Edmonton, AB T6G 2H7, Canada; esra@ualberta.ca; 2The Friends of Garrett Cumming Research & Muscular Dystrophy Canada HM Toupin Neurological Science Research Chair, 8613-114 Street, Edmonton, AB T6G 2H7, Canada

**Keywords:** CRISPR, gene editing, Duchenne muscular dystrophy (DMD), exon skipping, NHEJ, dystrophin

## Abstract

Duchenne muscular dystrophy (DMD) is an X-linked recessive neuromuscular disorder with a prevalence of approximately 1 in 3500–5000 males. DMD manifests as childhood-onset muscle degeneration, followed by loss of ambulation, cardiomyopathy, and death in early adulthood due to a lack of functional dystrophin protein. Out-of-frame mutations in the dystrophin gene are the most common underlying cause of DMD. Gene editing via the clustered regularly interspaced short palindromic repeats (CRISPR) system is a promising therapeutic for DMD, as it can permanently correct DMD mutations and thus restore the reading frame, allowing for the production of functional dystrophin. The specific mechanism of gene editing can vary based on a variety of factors such as the number of cuts generated by CRISPR, the presence of an exogenous DNA template, or the current cell cycle stage. CRISPR-mediated gene editing for DMD has been tested both in vitro and in vivo, with many of these studies discussed herein. Additionally, novel modifications to the CRISPR system such as base or prime editors allow for more precise gene editing. Despite recent advances, limitations remain including delivery efficiency, off-target mutagenesis, and long-term maintenance of dystrophin. Further studies focusing on safety and accuracy of the CRISPR system are necessary prior to clinical translation.

## 1. Introduction

Duchenne muscular dystrophy (DMD) is an X-linked recessive neuromuscular disorder characterized by severe muscle wasting, cardiomyopathy, and elevated creatinine kinase levels [1,2]. With a prevalence of ~1 in 3500–5000 males, it is the most common childhood muscular dystrophy [3,4]. Symptoms begin to manifest during early childhood, progressively worsening throughout adolescence with loss of ambulation and a resultant lifespan of only 20–40 years [5]. Although corticosteroids can be administered to aid in symptom management, there is currently no curative treatment for DMD [6]. 

DMD can be caused by a variety of different mutations in the *DMD* gene encoding dystrophin, which is the largest human gene spanning ~2.3 Mb on the X chromosome (Xp21.2-p21.1) [7,8,9,10]. Owing to its large size, there are multiple opportunities for mutations such as large deletions, point mutations, and duplications throughout the *DMD* gene [11]. The majority of these mutations disrupt the open reading frame (ORF), resulting in a frameshift and subsequently the production of a truncated, non-functional dystrophin protein [12]. Of note, in-frame mutations may lead to milder Becker muscular dystrophy (BMD) [13]. As the reading frame is maintained, a semi-functional dystrophin protein is expressed, allowing for some maintenance of muscle function. This has inspired therapeutic strategies including gene editing and exon skipping as a means of restoring the ORF in DMD patients. However, the location and size of the skipped region must be considered, as some dystrophin domains are vital to its function and cannot be perturbed.

### 1.1. Dystrophin Structure

Dystrophin acts as a tether between the actin cytoskeleton and the extracellular matrix in muscle cells, thereby providing membrane stability as a member of the dystrophin–glycoprotein complex (DGC) [14]. Accordingly, a lack of functional dystrophin protein results in fragile or damaged myofibers and subsequent muscle degeneration. At 427 kDa (3684 amino acids) in size, dystrophin is comprised of 4 different functional domains: an amino-terminal actin-binding domain, a central rod domain with 24 spectrin-like repeats, a cysteine-rich dystroglycan-binding domain, and a carboxy-terminal domain which interacts with the sarcolemma via dystrobrevin and syntrophin binding sites (Figure 1) [9,12]. The N and C termini are essential to maintain proper dystrophin function and membrane integrity. However, the central rod domain contains redundant repeats and thus can be shortened while still maintaining functionality [6]. Such internally truncated forms of dystrophin may be expressed in BMD patients, thereby providing evidence that skipping the associated exons is a promising treatment strategy for DMD patients [13]. Similarly, microdystrophin—a smaller version of the dystrophin gene intended for gene therapy—lacks the redundant rod repeats as it contains only the minimal functional components [15]. This further provides evidence that the N and C termini must be maintained during gene editing or exon skipping strategies, while a portion of the central domain is dispensable.

Certain regions of the dystrophin gene are more likely to be mutated in DMD, known as mutational “hotspots”. These include exons 45 to 55 (corresponding to the redundant rod domain), and exons 2–10 (corresponding to the actin-binding domain) (Figure 1) [16]. Emphasis tends to be placed on strategies that correct mutations in these exons, in order to increase the applicability to as many patients as possible.

### 1.2. Current Approved DMD Therapeutics

The current standard of care for DMD patients involves a multidisciplinary approach, including corticosteroids, physical therapy, and treatment of cardiomyopathy [17,18,19]. Corticosteroids such as prednisone and deflazacort have been demonstrated to improve overall muscle strength and preserve function, although they are not curative [17,20,21,22]. Additionally, uncertainty remains surrounding the ideal dosing strategy to balance treatment efficiency with side effect management. Due to these limitations, molecular therapies that target DMD at the source are appealing.

As BMD patients harboring in-frame *DMD* mutations often present with milder phenotypes, exon skipping has been proposed to restore the ORF and ameliorate some symptoms in DMD patients [13,23,24,25]. This can be achieved by antisense oligonucleotides (AONs): short, synthetic, single-stranded nucleic acid analogs that can bind to mRNA in a site-specific manner and modulate splicing. Currently, there are four FDA-approved AON treatments for DMD: eteplirsen (exon 51 skipping) [26], golodirsen (exon 53 skipping) [27], viltolarsen (exon 53 skipping) [28], and casimersen (exon 45 skipping) [29]. However, AONs must be repeatedly administered, as they only have a transient effect. In addition to repeated treatments and the associated high cost, AONs tend to have poor delivery and uptake into target tissues such as the heart [30]. AON therapy is also mutation-specific and therefore not applicable to all DMD patients. Overall, these drawbacks limit the efficacy of AONs in DMD [31]. In contrast to AONs, gene editing via CRISPR (clustered regularly interspaced short palindromic repeats) would be able to permanently correct the *DMD* gene, thereby abolishing the need for repeated treatments. CRISPR is therefore a promising therapeutic which could overcome many of the barriers faced by AONs.

### 1.3. CRISPR/Cas9 System

CRISPR technology has recently been explored as a gene-editing therapeutic for DMD. Originally identified as a bacterial defense system against viral pathogens, CRISPR has since been modified and exploited for gene editing [32,33]. The CRISPR/Cas9 system consists of two main components: a CRISPR-associated (Cas) endonuclease and a single guide RNA (sgRNA), which guides Cas9 to a specific 20-nucleotide site in the genome containing the complementary sequence. In the vicinity of an appropriate protospacer adjacent motif (PAM) sequence, the Cas9 endonuclease will generate targeted double-stranded DNA breaks (DSBs).

There are multiple different Cas endonucleases, each originating from a different bacterial species with a unique PAM sequence. The most commonly used Cas9 enzyme in CRISPR systems is SpCas9, from *Streptococcus pyogenes*, with a PAM sequence of 5′-NGG-3′ or 5′-NAG-3′ [32,33,34]. The PAM sequence for the *Staphylococcus aureus* Cas9 (SaCas9) is longer (5′-NNGRRT-3′), thereby limiting the flexibility of target sites [35,36,37]. However, SaCas9 boasts a smaller size; therefore, it is easier to package for viral delivery. CjCas9, from *Campylobacter jejuni*, is even smaller than SaCas9, but with a longer PAM sequence of 5′-NNNNACAC-3′ or 5′-NNNNRYAC-3′ [38]. In addition to naturally occurring Cas endonucleases, a few engineered varieties are available. For example, modifications have been made to SpCas9 to increase fidelity and reduce off-target effects. These modified Cas9 enzymes include eSpCas9(1.1) [39], SpCas9-HF1 [40], HypaCas9 [41], HeFSpCas9s [42], evoCas9 [43], xCas9 [44], HiFi Cas9 [45], Sniper-Cas9 [46], and SpCas9-NG [47]. Additionally, nickase Cas9 (nCas9) [48] and deactivated Cas9 (dCas9) [49] have been altered to reduce or abolish the endonuclease activity, respectively, thereby improving the safety of the CRISPR/Cas9 system as no DSBs are generated [50]. Instead, they can be fused to other components such as transcriptional activators [49,51] or precise base editors [52,53] for alternative therapeutic strategies.

Transcription activator-like effector nucleases (TALENs) and zinc-finger nucleases (ZFNs) have also been explored for gene editing in DMD [54,55,56,57]. Both of these technologies generate DSBs via a Fok1 endonuclease which can then be repaired in a variety of ways, similar to DSBs generated by CRISPR/Cas9. However, ZFNs lack targeting specificity and therefore may induce cytotoxic off-target effects [58,59,60]. While TALENs have improved specificity, their relatively large size makes packaging for in vivo delivery a significant challenge [55]. Thus, CRISPR/Cas9 remains the popular choice for gene editing in DMD therapeutics.

The CRISPR gene-editing system may be beneficial for other diseases beyond DMD. Since its first discovery in 1987, and subsequent first use in vivo in 2013 [61], CRISPR has been applied to numerous diseases and has even moved forward to clinical trials in some cases [62,63] (Figure 2).

### 1.4. Methods of Gene Editing Via CRISPR

DSBs generated by CRISPR/Cas9 can be repaired by one of two main endogenous repair pathways: non-homologous end joining (NHEJ) or homology-directed repair (HDR) (Figure 3) [64]. HDR requires an exogenous DNA template, which is then used for precise editing at the targeted site. In the context of DMD, the mutated exon(s) could be replaced via an exon “knock-in” approach, thereby restoring full-length dystrophin (Figure 3) [12]. While the accuracy of HDR is appealing, it occurs at low frequency in postmitotic cells such as skeletal muscle cells [12,64]. Additionally, there is a limit to the length of the template that can be delivered, thus it is not possible to correct large deletions spanning multiple exons. Due to these drawbacks, HDR is not often considered for DMD therapeutics. In postmitotic cells, the predominant DSB repair pathway is therefore NHEJ [64]. Described as error-prone in comparison to HDR, NHEJ generates small insertions or deletions (indels) at DSBs. These indels can restore dystrophin expression via multiple different mechanisms, including: i) exon skipping (single-cut), ii) exon deletion (double-cut), iii) exon reframing (single-cut) (Figure 3) [12,64,65,66]. First, an sgRNA targeted to an exon/intron junction can induce skipping of the out-of-frame exon (exon skipping). Next, by using two different sgRNAs flanking the exon(s) to be removed, the mutated exon(s) can be directly cut out, thereby restoring the ORF (exon deletion). Finally, random indel generation within an out-of-frame exon can restore the reading frame (exon reframing).

In this article, in vitro and in vivo preclinical applications of these three NHEJ methods will be discussed, in addition to other recent advancements in CRISPR therapeutics for DMD such as base editing, prime editing, and utrophin upregulation.

## 2. In Vitro and In Vivo Gene Editing Via CRISPR

The first proof-of-concept for in vivo CRISPR-mediated gene editing in DMD was performed by Long et al. in 2014 [67]. Using the established *mdx* mouse model harboring a nonsense mutation in *Dmd* exon 23 [68], Long et al. injected zygotes with SpCas9, sgRNA, and an exogenous DNA template for HDR [67]. The resultant mice were mosaic for corrected dystrophin, varying from 2 to 100% correction. As germline editing is not considered ethical in humans, focus has since shifted to postnatal gene editing instead. However, an important takeaway from this pilot study is that 100% editing efficiency is not required to restore wild-type (WT) dystrophin protein levels. In fact, Long et al. found that approximately 15% gene editing was sufficient.

### 2.1. Single-Cut Exon Skipping

The single-cut exon skipping method restores the dystrophin reading frame by disrupting an exon splice site, thus skipping the out-of-frame exon. The PAM sequence for SpCas9 is compatible with the universal splice acceptor site, allowing it to be readily targeted by the CRISPR system [12]. Additionally, it is estimated that >70% of DMD patients have mutations amenable to an exon-skipping approach [66,69]. Single-cut exon skipping also requires only one sgRNA as opposed to two for the double-cut approach, reducing the strain on the packaging and delivery system. Finally, as only one DSB is induced, this lowers the likelihood of gross DNA aberrations or deleterious mutations. This favourable mechanism of repair has therefore been tested in numerous in vitro and in vivo models.

In vitro, single-cut exon skipping has been used to skip exons within the exon 45–55 mutational hotspot. For example, Li et al. first performed this single-cut approach in DMD patient-derived induced pluripotent stem cells (iPSCs) with an exon 44 deletion (ΔEx44) [57]. By targeting SpCas9 to the exon 45 splice acceptor site, exon 43 was subsequently directly joined with exon 46, allowing for an in-frame dystrophin transcript. Similarly, Long et al. used a single-cut approach to skip exon 51 in ΔEx48-50 patient iPSCs [70]. In addition to dystrophin protein restoration, Long et al. assessed functional improvements in the force of contraction generated from engineered heart muscle derived from edited cardiomyocytes. Only 30–50% of the cardiomyocytes needed to be corrected to improve muscle contraction. This result echoes previous findings, indicating that 100% gene editing efficiency is unnecessary to ameliorate some symptoms of DMD [67,70].

The single-cut exon-skipping strategy has also been tested in vivo, in both mouse [71,72,73] and canine models [74]. In DMD patients, an exon 50 deletion leading to an out-of-frame *DMD* transcript is one of the most common single exon deletions [11,71]. Accordingly, exon 51 skipping via splice site disruption is a promising therapeutic strategy for these patients. To test this in vivo, CRISPR/Cas9 was used to generate a new DMD mouse model, harboring an exon 50 deletion (ΔEx50) [71]. These mice were subsequently treated with a single-cut exon-skipping strategy targeted to exon 51, leading to efficient restoration (up to 90%) of dystrophin expression in skeletal and cardiac muscle. This strategy has also been deployed in a ΔEx50 canine model of DMD (deltaE50-MD) [74]. CRISPR/Cas9 components were administered intramuscularly with sgRNAs targeted to exon 51 for either NHEJ reframing via indel generation or single-cut exon skipping. Up to 90% dystrophin restoration was observed by Western blotting, and no adverse effects such as an immune response, off-target mutagenesis, or liver toxicity were found. With a sample size of only two dogs, this landmark study bears some limitations and further work is required to fully investigate the long-term effects of CRISPR/Cas9 therapy in a large mammal model. Both of these studies are highly relevant to treating DMD patients, due to the exon 50 deletion as opposed to a nonsense mutation in the oft-used *mdx* mouse model [68,71]. Additionally, these studies used adeno-associated virus (AAV9) vectors, as is common practice for the systemic delivery of CRISPR components in vivo due to known tropism for skeletal and cardiac muscle [75].

In addition to rectifying exon 50 deletion mutations, CRISPR/Cas9 has been demonstrated to restore the reading frame in exon 44 deletion models [72]. Therapeutics which correct exon 44 deletions could treat ~12% of DMD patients, as it is one of the most common mutations. Using both patient-derived iPSCs (ΔEx44) and a CRISPR-generated ΔEx44 mouse model, Min et al. effectively skipped exon 45, allowing exon 43 to join directly to exon 46 and thus restoring the *DMD* reading frame [72]. In the corrected ΔEx44 mouse model, 90–95% of myofibers from various skeletal and cardiac muscles exhibited dystrophin expression by immunostaining. Similar results were seen for the correction of other exon deletions within the mutational hotspot, including exons 43, 45, and 52 [76].

### 2.2. Double-Cut Exon Deletion

Contrary to single-cut exon skipping, double-cut exon deletion requires two sgRNAs flanking either side of the exon to be directly cut out. Notably, this method applies to a larger proportion of DMD patients as it is less mutation-specific, allowing for the removal of multiple exons in a mutational hotspot. However, this may lead to a shorter dystrophin protein as compared to the single-cut exon-skipping approach. The implications of this will vary based on the specific targeted region; therefore, careful consideration is required to avoid perturbing essential binding domains. The use of two distinct sgRNAs also increases the possibility of deleterious off-target effects, and generating two simultaneous DSBs comes with the risk of unwanted damaging mutations or rearrangements. Additionally, the efficiency of a double-cut approach is lower than single-cut, as two DSBs must take place simultaneously for exon deletion to occur. Finally, delivering two distinct sgRNAs adds an additional challenge to the delivery process. Nonetheless, the appeal of increased applicability ensures that the double-cut exon deletion approach is still widely tested in vitro and in vivo.

The exon 45–55 mutational hotspot is an appealing target for a multi-exon deletion strategy [77]. This ΔEx45-55 deletion has been associated with BMD and remarkably mild symptoms, thus it is a promising treatment strategy as it has the potential to correct ~60% of *DMD* deletion mutations [77,78]. When tested in ΔEx48-50 DMD patient-derived myoblasts, Ousterout et al. found that deleting the entire exon 45–55 region restored dystrophin expression as confirmed by Western blot [77]. However, it was noted that this was less efficient than deleting exon 51 alone, indicating that editing efficiency decreases in a size-dependent manner. Accordingly, a balance must be struck between editing efficiency and the applicability advantage of targeting multiple exons. This same region was targeted by Young et al. in DMD patient-derived iPSCs, resulting in stable dystrophin protein expression and improved membrane stability [79]. Other large double-cut deletions involving this hotspot have also been tested, including exons 44–54 [80] and 48–57 [81].

As mentioned previously, there is an additional mutational hotspot comprising of the N-terminal actin-binding domain. However, because this domain has functional importance, large multi-exon deletions of this region may produce unstable or non-functional dystrophin proteins. Kyrychenko et al. tested three multi-exon deletions within this hotspot (exons 3–9, 6–9, or 7–11) in iPSCs, taking care to ensure the preservation of at least one actin-binding site following the deletion [82]. While all three deletions restored the dystrophin reading frame, the ΔEx3-9 strategy resulted in the greatest functional improvement in iPSC-derived cardiomyocytes. This study demonstrated that CRISPR can be utilized in the N- or C-terminal functional domains, as long as particular amino acid residues are preserved to retain functionality.

In addition to multi-exon deletions, single-exon deletions via the double-cut strategy have been tested in vitro [70,80,83,84,85]. Although deleting a smaller region lessens some of the challenges associated with the double-cut strategy, it also abolishes the applicability advantage of a multi-exon deletion.

In vivo, single-exon deletion via the double-cut strategy has been demonstrated numerous times. In 2016, three separate studies used CRISPR to effectively delete *Dmd* exon 23 in the *mdx* mouse model [86,87,88]. Following systemic delivery via AAV vectors, dystrophin expression was found to be partially restored with proper localization. Recently, an additional study involving the excision of *Dmd* exon 23 in *mdx* mice was performed, to assess long-term dystrophin expression [89]. Dystrophin was still detected in cardiac and skeletal muscle via Western blot and immunostaining 1-year post-treatment. In addition to mouse models, double-cut single exon deletion has been tested in a pig model of DMD (ΔEx52) [85,90]. Following AAV9-mediated delivery, exon 51 was excised, successfully restoring widespread dystrophin expression [85].

Double-cut multi-exon deletion has also been tested in vivo. For example, Young et al. deleted exons 45–55 in a novel humanized DMD (hDMD) del45/*mdx* mouse model, which contains a human *DMD* transgene (ΔEx45) in an *mdx* background [91]. The large deletion was successful and dystrophin-positive muscle fibers were observed via immunostaining. Additionally, Bengtsson et al. deleted exons 52–53 to successfully restore the reading frame in *mdx^4cv^* mice, harboring a nonsense mutation in exon 53 [92].

### 2.3. Single-Cut Exon Reframing

Single-cut exon reframing attempts to restore the ORF by generating small indels. While this would preserve a relatively larger dystrophin protein as compared to the other NHEJ-based methods, there is theoretically only a 1/3 chance of restoring the reading frame when an indel is generated. While it has been tested both in vitro [57,77,80,83,93] and in vivo [71,72,74,76,83,94,95,96], reframing may be inconsistent [80]. To combat the sporadic nature of single-cut exon reframing, Min et al. designed a sgRNA that results in a single adenosine insertion at the DSB due to a single nucleotide overhang [72]. Accordingly, this method could be used to consistently restore the ORF in exons that are off by only one nucleotide. Additional advancements in single-cut reframing have recently been made, both in vitro and in vivo [72,76]. Similar to single-cut exon skipping, exon reframing requires only one sgRNA and subsequently one DSB location. This has numerous benefits over the double-cut strategy such as a reduced opportunity for genotoxic rearrangements and higher editing efficiency.

## 3. Novel Developments

Beyond the traditional NHEJ-based CRISPR/Cas9 gene editing, numerous modifications have been made to modify the editing outcome or enhance efficiency. For example, deactivating the endonuclease component of CRISPR/Cas9 (nCas9, dCas9) allows for the generation of a precise, targeted enzyme that can be conjugated to different proteins such as transcriptional activators or base editors [48,49,50]. Novel advancements such as these will be discussed in the following subsections.

### 3.1. Base Editing

In order to circumvent some of the risks associated with DSBs, a deactivated Cas9 protein (dCas9) has been generated with no endonuclease activity. Upon fusing to a cytosine or adenosine deaminase, the dCas9 system can perform precise base editing (C:G > T:A or A:T > G:C, respectively) as it does not rely on error-prone NHEJ repair pathways (Figure 4A) [53,97]. These base editors can be deployed to repair a nonsense mutation directly [98,99], or induce exon skipping by altering the sequence at a splice site [52,100]. As approximately 25–35% of DMD patients carry point mutations in the dystrophin gene, base editing is a promising therapeutic method [11,12]. Additionally, correcting a point mutation would allow for restoration of the entire full-length dystrophin protein.

Of note, precise base editing with an adenosine deaminase was performed in a mouse model of DMD with a nonsense mutation in exon 20 [98]. By converting the premature stop codon (TAG) to a glutamine codon (CAG), dystrophin expression was restored in ~17% of myofibers, with no sporadic indels nor off-target effects. A similar approach was taken recently to correct the *mdx^4cv^* mouse model [92]. Following systemic administration of an adenosine base editor, widespread dystrophin restoration was seen in skeletal and cardiac muscle, along with the correction of over 95% of cardiomyocytes [99]. This study demonstrated the potential for continued improvements in CRISPR/Cas9 efficiency and specificity.

In addition to correcting point mutations, base editors can also be used to skip entire exons, similar to the NHEJ-mediated single-cut exon skipping. As base editors do not induce DSBs, they may be preferable from a safety and efficiency standpoint. Additionally, skipping an entire exon expands the applicability of the base editing strategy as compared to point mutation correction. By targeting the 5′ splice site with a CRISPR-guided cytidine-deaminase, Yuan et al. successfully skipped exon 50 in DMD patient-derived iPSCs (ΔEx51) [52]. Upon differentiation to cardiomyocytes, ~90% of the DNA was found to be edited allowing for phenotypic improvements such as reduced creatine kinase levels.

While base editors allow for precise correction and thus may be preferable to traditional NHEJ CRISPR/Cas9 strategies, the size constraints of AAV-mediated delivery limit the translational applications due to the large size of the dCas9-deaminase fusion protein. Novel delivery strategies must therefore be explored, including trans-splicing AAV vector systems wherein the base editor system is delivered as gene fragments in two separate vectors [98,101,102].

### 3.2. Prime Editing

Prime editing is a recent and promising addition to the CRISPR/Cas9 gene-editing system [103]. Similar to precise base editing, prime editing takes advantage of a catalytically inactive nCas9, therefore no DSBs are generated. Instead, the nCas9 is fused to a modified reverse transcriptase and delivered with a “prime editing” guide RNA (pegRNA). The reverse transcriptase component can copy the DNA template from the pegRNA at the site of the nicked DNA strand. The pegRNA thereby acts as a donor template for precise gene repair via base pair conversions, insertions, or deletions [104]. Prime editing is not limited to only C:G > T:A or A:T > G:C conversions, unlike base editors. While prime editing has the potential to correct a variety of DMD-causing mutations, size constraints remain an issue for in vivo delivery.

Recently, Chemello et al. were the first to test prime editing in the context of DMD [105]. Using DMD patient-derived iPSCs (ΔEx51), they successfully reframed exon 52 and restored dystrophin expression to 24.8–39.7% of WT levels. Contractile function also exhibited improvements for cardiomyocytes differentiated from the edited iPSCs. This proof-of-concept study has demonstrated the applicability of prime editing to DMD, although further testing in vivo is required.

### 3.3. Utrophin Upregulation

Utrophin is a dystrophin homolog that may partially compensate for a lack of dystrophin protein in DMD. While it plays a membrane-stabilizing role in fetal muscle cells, utrophin is normally replaced by dystrophin by adulthood [65,106]. Utrophin upregulation may therefore help maintain muscle cell integrity and alleviate some of the pathologies of DMD. Through a dCas9-transcriptional activator fusion protein, the *UTRN* promoters can be targeted thus increasing expression (Figure 4B) [49]. This was demonstrated by Wojtal et al. in DMD patient-derived myoblasts (ΔEx42-52), resulting in a 1.7–6.9-fold increase in utrophin expression [51]. Utrophin upregulation is an appealing therapeutic target as it is not *DMD* mutation-specific and thus applicable to all patients. Additionally, no DSBs are generated, therefore increasing the safety of the approach. However, long-term functional benefits of utrophin overexpression are unknown, and the large size of the fusion protein poses challenges for AAV-mediated delivery.

Increased utrophin expression was achieved through a different approach by Sengupta et al., negating the need for a transcriptional activator fusion protein [107]. CRISPR/Cas9-mediated genome editing was used to abolish five microRNA (miRNA) binding sites within the *UTRN* 3′ untranslated region that normally repress utrophin expression. By deleting these sites, utrophin proteins levels were found to be two-fold higher in DMD patient-derived iPSCs. Functional improvements were also seen in myotubes differentiated from the edited iPSCs.

### 3.4. Homology-Independent Targeted Integration (HITI)

As previously discussed, exon knock-in via HDR is not typically suitable for restoring dystrophin expression in DMD due to its low efficiency in postmitotic cells [64]. However, the theory behind this approach is appealing as it allows for the restoration of a full-length dystrophin protein, unlike exon skipping or deletion strategies. This is of particular importance when considering mutations in the N- or C-terminal regions of dystrophin as these bear functional importance and thus cannot tolerate CRISPR-mediated deletions [9]. A novel exon knock-in method with relatively high efficiency in postmitotic cells was recently reported, termed homology-independent targeted integration (HITI) [108]. HITI involves the delivery of a donor plasmid, which includes two Cas9 cleavage sites flanking the intended donor sequence (Figure 4C). Once Cas9 cleaves both the donor plasmid and the targeted genomic DNA, the NHEJ repair system will integrate the donor sequence. As HITI relies on NHEJ, it can occur irrespective of the cell cycle state, unlike traditional HDR [109]. While this is a promising exon knock-in approach to restore full-length dystrophin protein, it has yet to be tested in the context of DMD.

### 3.5. Single-sgRNA Correction of Duplication Mutations

Single or multi-exon duplications account for up to 10–15% of DMD patient mutations, yet they are difficult to study in vivo due to a lack of appropriate animal models [11]. Recently, a mouse model with a head-to-tail duplication of exons 18–30 was generated with CRISPR/Cas9 [110]. This mutation was then rectified with a single-sgRNA gene-editing strategy targeting intron 21 of the *Dmd* gene: both copies of the duplicated region were cut by Cas9, resulting in the removal of the region between the sgRNAs upon DNA repair and restoration of the *Dmd* ORF. With ~4–18% dystrophin protein restoration across various muscle types, improvements were observed in muscle pathology, strength, and locomotor function. This single-sgRNA strategy boasts many advantages such as packaging efficiency and restoration of full-length dystrophin protein. While still in early development, it shows promise for correcting duplication mutations in DMD and other duplication disorders.

## 4. Challenges and Future Directions

While there have been numerous advancements in CRISPR/Cas9 therapeutics, many challenges remain. Accordingly, additional studies are necessary prior to clinical translation.

### 4.1. Delivery

One key challenge for CRISPR therapeutics is the delivery method, particularly when considering large Cas9 fusion proteins in the case of base or prime editors. The CRISPR/Cas9 complex is often delivered using viral vectors such as AAVs, which have an inherent packaging limit of ~4.7 kb [111]. SpCas9 alone (~4 kb) already approaches this limit, thereby the addition of more components such as sgRNAs, a transcriptional activator, or a reverse transcriptase proves impractical with AAV-mediated delivery. To circumvent this challenge, multiple AAV vectors can be used to deliver separate components of the CRISPR/Cas9 system, or a smaller Cas9 protein (ex. SaCas9) can be used. In addition to the size constraints, viral vectors may cause an immune response due to pre-existing antibodies against AAV [112]. This poses further challenges as the CRISPR/Cas9 system must be administered in a single tolerable dose, with no possibility for a second dose due to high antibody titers generated during the first treatment. Another challenge associated with AAV-mediated delivery is the risk of AAV integration into DNA DSB sites [113]. Finally, AAV vectors may accumulate in the liver, posing the risk of deleterious side effects or dose-dependent toxicity [6].

As an alternative to AAV-mediated delivery, lipid or gold nanoparticles are being explored [73,114,115]. Nanoparticles can be taken up via endocytosis and boast many advantages such as reduced off-target mutagenesis, low cost, and decreased immunogenicity. However, efficient systemic delivery remains a challenge. For example, Lee et al. found only 1% gene editing efficiency when using nanoparticles [115]. Similarly, Wei et al. restored dystrophin protein levels up to 4.2% following delivery with lipid nanoparticles [73]. Improvements in delivery efficiency must be made prior to the widespread use of engineered nanoparticles.

### 4.2. Immunogenicity

In addition to the potential immunogenicity associated with AAV vectors, other components of the CRISPR/Cas9 may induce an immune response. First, as the Cas9 enzyme is bacterial in origin, it will likely elicit an immune reaction [116]. This could be alleviated using modified Cas9 proteins, wherein particular epitopes have been modified to reduce recognition by the immune system [117]. While there is also a concern of the restored dystrophin protein eliciting an immune response, DMD patients have a low frequency of spontaneous “revertant fibers”, wherein the reading frame is maintained [118]. Accordingly, dystrophin should not be recognized as foreign to the immune system. Finally, sgRNAs have been found to initiate an innate immune response in some conditions [119,120]. However, as many DMD patients are already administered corticosteroids as an anti-inflammatory treatment, this may suppress any mild immune response and thus render the issue nonsignificant.

### 4.3. Off-Target Mutagenesis

A prominent safety concern for CRISPR/Cas9 gene editing is the possibility of off-target mutagenesis [121,122,123,124]. Despite careful sgRNA design, the CRISPR/Cas9 system can tolerate a few mismatches and thus has the potential to generate DSBs at unintended sites. Despite this risk, off-target editing appears to occur at a low rate (<1%) in postmitotic skeletal muscle cells [6]. The risk of mutagenesis can be further minimized by several different strategies. First, modified high-fidelity Cas9 proteins such as eSpCas9(1.1) [39] or SpCas9-HF1 [40] can be used in place of traditional Cas9. A paired nicking strategy could also be used, with nCas9 generating offset single-stranded breaks as opposed to DSBs [48]. Off-target single-stranded breaks can be repaired by base-excision repair, a high-fidelity DNA repair system, thereby decreasing off-target mutagenesis. Additionally, muscle-specific promoters could spatially limit the expression of CRISPR/Cas9 components, thus reducing the likelihood of off-target effects in other tissues [71,96]. Finally, careful sgRNA design can aid in increasing the specificity of cut sites [125]. In silico programs such as Cas-OFFinder [126] and CRISPRscan [127] can be used to design efficient sgRNAs with low off-target effects.

### 4.4. Long-Term Efficacy

Long-term maintenance of the edited *DMD* gene is also a potential challenge, due to the natural turnover of skeletal muscle cells [12]. Satellite cells are typically not edited by CRISPR/Cas9, therefore the edited nuclei may be diluted as new muscle cells are generated [128]. Additionally, due to antibodies generated against AAV, CRISPR/Cas9 treatment can only be administered once. This adds to the challenge of long-term efficacy as the treatment must be durable, with no possibility of a second dose later in life.

Recently, the long-term maintenance of dystrophin protein restoration was investigated in a mouse model of DMD [129]. While dystrophin persisted in cardiomyocytes, it was lost in skeletal muscle fibers due to higher turnover. To circumvent this challenge, Bengtsson et al. co-delivered a microdystrophin transgene along with the CRISPR gene editing components. The microdystrophin transgene effectively stabilized the myofibers, slowing the degeneration–regeneration cycle and maintaining dystrophin expression.

### 4.5. Maintaining Dystrophin Structure

As previously discussed, exons cannot be skipped haphazardly when attempting to restore the *DMD* ORF [16]. Instead, certain functional domains or binding sites must be conserved, while other redundant domains can readily be skipped or deleted [9]. This is to ensure that dystrophin is still able to join the DGC, acting to stabilize muscle cell membranes. Additionally, certain methods of CRISPR/Cas9 gene editing are superior for conserving the largest possible dystrophin protein. For example, HITI exon knock-in [108], NHEJ reframing [71,72,74,77,80,83,96], precise base editing [53,97,98,99], and prime editing [103,104,105] are all able to restore full or nearly full-length dystrophin proteins. On the other hand, single-cut exon skipping or double-cut exon deletion via NHEJ lead to a shorter form of dystrophin. While such internally truncated dystrophin proteins may retain basic functionality, preserving more of the full-length sequence is preferable when possible.

### 4.6. Applicability

While CRISPR/Cas9-mediated exon skipping strategies are theoretically applicable to ~70% of DMD patients, there are still some mutations that cannot be rectified with gene editing [66,69]. For example, excessively large deletions of multiple exons, or deletions in the N- or C- termini cannot be repaired with any of the current strategies. Additionally, single-cut exon skipping via NHEJ is mutation-specific, and thus not generalizable. Double-cut exon deletion via NHEJ may help alleviate this challenge.

### 4.7. Comparison with Other Gene Corrective Approaches

Cell transplantation or gene replacement therapy with microdystrophin may be an alternative solution for patients with mutations not amenable to CRISPR gene editing [15,130,131,132]. Cell transplantation involves the delivery of stem cells, induced pluripotent stem cells, or myoblasts, in an attempt to replace a patient’s own deteriorating cells [130]. Either donor or autologous cells may be used, although the latter must be repaired or edited prior to reintroduction. While this approach has had some success in vivo, many limitations remain such as limited cell numbers, immune response to transplantation, low cell migration ability, and potential tumour formation in the case of stem cells [130,132,133,134]. Microdystrophin gene therapy is also a potential alternative to gene editing [15,131,135]. This approach involves the delivery of an exogenous, truncated form of the dystrophin gene. The redundant rod domain is shortened, leaving the minimum amount of protein required to maintain proper dystrophin function. Following success in animal models, multiple in-human clinical trials were initiated [136,137,138,139,140]. However, concerns regarding patient safety have arisen following serious adverse events, including one death [141]. In addition to these side effects, gene therapy comes with a high cost, as well as the potential for an immune response against either the viral delivery vector or dystrophin itself [135]. Overall, while both of these approaches circumvent the applicability limitations of CRISPR gene editing, further development is required to limit detrimental side effects. 

## 5. Conclusions

Overall, CRISPR-mediated gene editing remains a promising therapeutic for the correction of *DMD* mutations. The advantages of permanent gene editing are numerous, as this would treat DMD at the source as opposed to merely managing symptoms through corticosteroids or physical therapy. CRISPR therapeutics would only require a single treatment, compared to multiple yearly administrations as is the case for antisense oligonucleotide therapy. Additionally, correcting the endogenous *DMD* gene allows for correct spatial and temporal regulation of dystrophin expression—unlike the microdystrophin transgene. Though many challenges remain, great improvements in CRISPR efficacy, fidelity, and safety have been made in the past few years. Notably, the advent of precise editing via base or prime editors increases the feasibility of CRISPR as a reliable therapeutic. Further in vivo studies investigating the long-term effects of *DMD* gene editing are still required prior to clinical translation. Additionally, improved delivery systems that increase efficiency and nullify packaging limitations are a critical next step. If CRISPR therapeutics are translated into clinical use for DMD, this would open the door for permanently correcting other monogenic disorders.

## Figures and Tables

**Figure 1 ijms-23-01832-f001:**
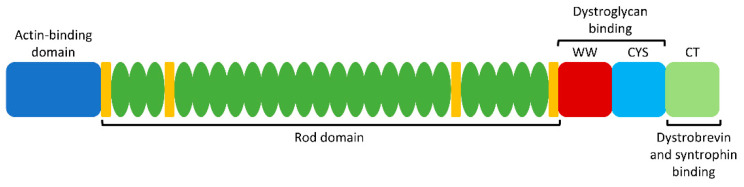
Dystrophin protein structure. Key functional domains, from left to right: N-terminal actin-binding domain, central rod domain with 24 repeats and 4 hinge regions, WW and cysteine-rich (CYS) domains (dystroglycan binding site), and carboxy-terminal (CT) domain (dystrobrevin and syntrophin binding sites). Since the rod domain is partially redundant, skipping or removing part of this region is typically well tolerated. Conversely, the N and C termini are essential for proper dystrophin function.

**Figure 2 ijms-23-01832-f002:**
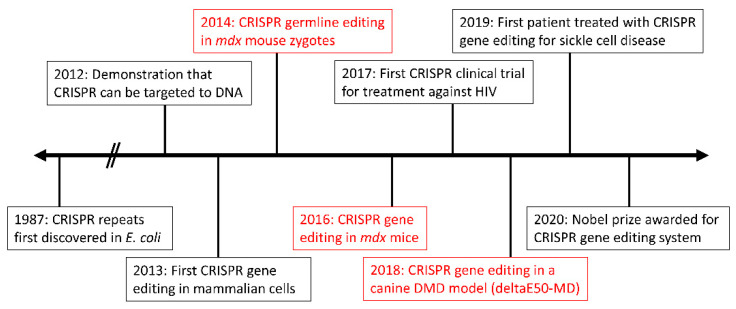
Timeline highlighting the major milestones in CRISPR gene editing for human diseases. Achievements in the field of DMD research are in red.

**Figure 3 ijms-23-01832-f003:**
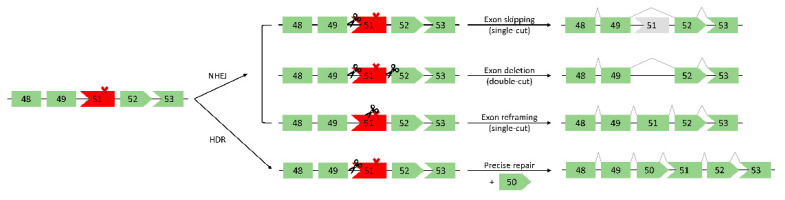
A schematic of DNA repair systems for gene editing with CRISPR/Cas9 in a hypothetical DMD patient harboring an exon 50 deletion mutation. Deleting exon 50 abolishes the reading frame and leads to a premature stop codon in exon 51, denoted by the “X”. Non-homologous end joining (NHEJ) or homology-directed repair (HDR) can restore the dystrophin reading frame. NHEJ exon skipping (single-cut) targets a splice site, thus skipping the adjacent exon. NHEJ exon deletion (double-cut) requires two sgRNAs flanking the out-of-frame exon(s) for removal. Exon reframing (single-cut) relies on indels generated by NHEJ to reframe the out-of-frame exon. Finally, HDR requires an exogenous DNA template to replace the missing or mutated exon in a precise manner. Scissors represent the sites targeted by CRISPR/Cas9.

**Figure 4 ijms-23-01832-f004:**
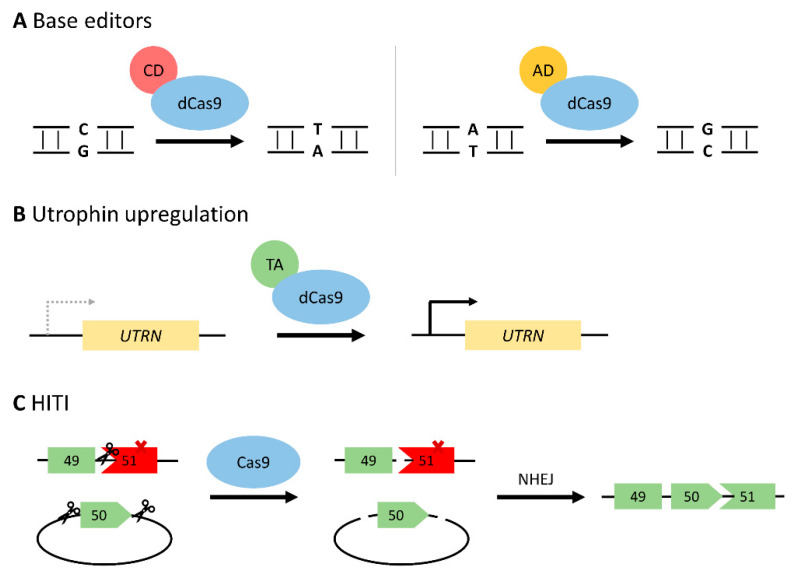
Novel developments to the CRISPR/Cas9 system. (**A**) Precise base editing via dCas9 fused to a cytosine (C:G > T:A) or adenosine (A:T > G:C) deaminase. Base editing could either repair a nonsense mutation, or induce exon skipping by targeting the splice site. (**B**) Transcriptional activator targeted to the utrophin promoter (*UTRN*) in an attempt to compensate for the loss of functional dystrophin. (**C**) Homology-independent targeted integration to knock-in exon 50 in a hypothetical DMD patient lacking exon 50, leading to a frameshift and premature stop codon in exon 51 (denoted by X). A donor plasmid is delivered with the desired exon, flanked by CRISPR/Cas9 cut sites. Cas9 will cleave both the genomic DNA and the donor plasmid, followed by NHEJ. Scissors represent Cas9 cut sites. Abbreviations: dCas9, deactivated Cas9; CD, cytosine deaminase; AD, adenosine deaminase; TA, transcriptional activator; HITI, homology-independent targeted integration; NHEJ, non-homologous end joining.

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
