# Peer review of "CRISPR Therapeutics for Duchenne Muscular Dystrophy"

_ijms, 2022, doi:10.3390/ijms23031832_

Round 1

Reviewer 1 Report

The use of CRISPR technology to cure Duchenne muscular dystrophy (DMD) has been discussed by the authors. DMD is caused by mutations in the dystrophin gene. One of the most promising treatments for DMD patients is gene editing using the CRISPR system.

Minor revision:

  1. Could you please include a figure (could be figure-1) for the Dystrophin structure and a summary of the dystrophin mutation?
  2. What are your thoughts on the success rate of novel CRISPR technology modifications such as base and prime editors? please add one paragraph with relevant references.
  3. Please include a schematic diagram (could be figure-3) based on your explanations in section 3 (Novel developments).
  4. Because the authors used the same heading in the 1.1 section, please modify the heading at 4.5 Dystrophin structure.
  5. Add more relevant references to section 4.5.

Author Response

Reviewer 1:

Minor revision:

Could you please include a figure (could be figure-1) for the Dystrophin structure and a summary of the dystrophin mutation?

response

Figure 1 has been added, outlining the structure of dystrophin and key functional domains.

What are your thoughts on the success rate of novel CRISPR technology modifications such as base and prime editors? please add one paragraph with relevant references.

response

Thank you for the important comments. While the base and prime editors are promising approaches, we feel that it is too early to predict the success rate of these technologies at this point as these are still at the very early stages.

Please include a schematic diagram (could be figure-3) based on your explanations in section 3 (Novel developments).

response

A schematic describing these novel developments has been added to section 3.

Because the authors used the same heading in the 1.1 section, please modify the heading at 4.5 Dystrophin structure.

response

Section 4.5 has been changed to “Maintaining dystrophin structure”.

Add more relevant references to section 4.5.

response

References have been added to section 4.5. These do not show as “tracked changes” due to the citation manager, but references were added for examples of primary literature for each of the methods proposed for maintaining full or nearly full-length dystrophin.

Reviewer 2 Report

Erkut and Yokota are providing a very interesting review that retraces the salient literature of CRISPR-based gene editing to treat DMD. The review is very well written, salient literature reports are properly cited and this referee supports publication in this journal. The referee suggests to update the final version of the manuscript with an additional figure to help readers to digest the huge amount of information contained in the review.

Minor

-Line 35. Please remove the term “completely”. The term may cause misleading interpretations to readers.

-“Current approved DMD therapeutics” is a broad paragraph title that should include all the approaches used to treat and/or mitigate DMD. Unfortunately, the paragraph is digested fast, limited to AOs and do not report the use of corticosteroids as commonly used agents.

-A figure reporting a time line with the major milestones achieved in the field of CRISPR-based gene therapy should be included. The figure should also emphasize the major achievements that CRISPR has had in the field of DMD

-A paragraph discussing the pro and drawbacks of CRISPR in comparison to other gene corrective approaches (genes and cell-based) should be contemplated and discussed in the manuscript.

Author Response

Reviewer 2:

-Line 35. Please remove the term “completely”. The term may cause misleading interpretations to readers.

response

The term has been removed.

-“Current approved DMD therapeutics” is a broad paragraph title that should include all the approaches used to treat and/or mitigate DMD. Unfortunately, the paragraph is digested fast, limited to AOs and do not report the use of corticosteroids as commonly used agents.

response

An additional paragraph has been added, briefly addressing corticosteroid use. A detailed three-part current standard of care has been referenced (Birnkrant et al. 2018), should readers want a more in-depth report on clinical DMD management. Emphasis is placed on molecular therapies (AONs) as these strategies are most relevant to this review.

-A figure reporting a time line with the major milestones achieved in the field of CRISPR-based gene therapy should be included. The figure should also emphasize the major achievements that CRISPR has had in the field of DMD

response

This has been added as figure 2.

-A paragraph discussing the pro and drawbacks of CRISPR in comparison to other gene corrective approaches (genes and cell-based) should be contemplated and discussed in the manuscript

response

A paragraph discussing microdystrophin gene therapy and cell-based therapies has been added to section 4.6.